# *Hermetia illucens* Fermented with *Lactobacillus plantarum* KCCM12757P Alleviates Dextran Sodium Sulfate-Induced Colitis in Mice

**DOI:** 10.3390/antiox12101822

**Published:** 2023-10-01

**Authors:** Seok Jun Son, Ah-Ram Han, Mi Jeong Sung, Sun Mee Hong, Sang-Hee Lee

**Affiliations:** 1Korea Food Research Institute, Iseo-myeon, Wanju-Gun 55365, Jeollabuk-do, Republic of Korea; ssj3203@kfri.re.kr (S.J.S.); lambo3@kfri.re.kr (A.-R.H.); dulle5@kfri.re.kr (M.J.S.); 2Department of Technology Development, Marine Industry Research Institute for East Sea Rim, Jukbyeon, Uljin-gun 36315, Gyeongsangbuk-do, Republic of Korea; hongsunmee@mire.re.kr

**Keywords:** *Hermetia illucens*, *Lactobacillus plantarum*, riboflavin, inflammatory bowel disease, microbiota, gene expression

## Abstract

Inflammatory bowel disease (IBD) can severely affect humans and animals and is difficult to treat. Black soldier fly (*Hermetia illucens*; Hi) larvae (BSFL) are a sustainable source of protein. However, no studies exist on the antioxidant and anti-inflammatory functions of BSFL or fermented BSFL with respect to IBD. In this study, riboflavin-producing *Lactobacillus plantarum* KCCM12757P was isolated from a fish farm tank, and in conjunction with hot water-extracted Hi (HeHi) (termed HeHi_Lp), was used to determine optimal fermentation conditions to increase vitamin B_2_ concentration. This in vivo study investigated the therapeutic effects and mechanistic role of HeHi_Lp in chronic colitis-induced murine models. Histological changes, inflammatory cytokine levels, and intestinal barrier function were explored. Gut microbial communities and gene expression in the nuclear factor (NF)-κB signaling pathway were also studied. HeHi_Lp remarkably reduced the disease activity index, inflammatory cytokine (inducible nitric oxide synthase, cyclooxygenase 2, tumor necrosis factor α, interleukin (IL-6 and IL-1β) levels, and increased body weight and colon length. HeHi_Lp administration significantly raised zonula occludens 1, occludin and claudin 1 and improved the composition of the gut microbiota and beneficial intestinal bacteria. These results suggest that HeHi_Lp can be used as a dietary supplement in pet food to alleviate colitis.

## 1. Introduction

Inflammatory bowel disease (IBD), which is mostly characterized by Crohn’s disease and ulcerative colitis, is a chronic inflammatory intestinal disease [1]. All types of IBD usually involve severe diarrhea, abdominal pain, rectal bleeding, fatigue, weight loss, weakening, and sometimes development of life-threatening complications [2]. IBD is prevalent not only in humans but also in pets such as dogs and cats. Canine and feline IBD is a group of chronic enteropathies characterized by repetitive or continuous gastric symptoms with an undiscovered etiology related to histopathological changes in the mucosa of the intestinal tract [3]. Diverse methods are available for studying IBD in animal models of colonic inflammatory injury. Typically, IBD models are classified into five groups: chemically induced, cell transfer, spontaneous, genetically engineered, and congenital [4]. Among the several mouse colitis models, the dextran sodium sulfate (DSS)-induced colitis model is widely used because of its simplicity and time- and cost-effectiveness [4,5]. Therefore, DSS is widely used to investigate the pathogenesis of IBD.

The black soldier fly (BSF), *Hermetia illucens* (Hi), is a true Diptera of the family *Stratiomyidae* [6]. BSF larvae (BSFL) feed on diverse organic materials and have been used for waste management purposes using substances such as rice straw, muck, distillers’ grains, food waste, and feces sludge [7,8,9,10,11,12]. BSFL diet and oil are considered alternatives to animal feeds because of their high protein concentrations even when fed with plant-based waste streams [13,14,15]. BSFL are typically high in both protein and fat, with protein content of 30–53 g/100 g dry matter (DM), lipid content of 20–41 g/100 g DM, and chitin, which predominately acts as a fiber in the human body [16]. They also contain adequate amounts of other essential minerals and vitamins at levels equivalent to or higher than those in other insects. Mineral and vitamin analysis showed that BSFL contained 15% ash, 0.56% manganese, 3.07% sodium, 0.57% iron, 2.27% potassium, 0.24 mg/100 g thiamin and 1.3 mg/100 g vitamin E [17,18]. However, as in the case of other insects, sensory and cultural issues prevent their use as human food; hence currently, they are only used as animal feed.

In 1998, probiotics were defined by the World Health Organization (WHO) and Food and Agriculture Organization as “live microorganisms that, when consumed in adequate amounts, can confer health benefits on the hosts” [19]. Various benefits of probiotics and their potential use in the treatment and control of IBD have been reported [20,21]. However, the mechanisms by which probiotics protect against IBD remain inconclusive. The gut microbiota contains an abundance of commensal *Lactobacillus* species that can restore homeostasis in the intestines under unstable conditions, and hence protect against IBD [22,23]. Probiotics can also strengthen the gut barrier function by regulating the production of cytokines, stimulating regulatory T cells, and aiding the survival of intestinal epithelium cells [24]. Certain strains of probiotic lactic acid bacteria (LAB) exert beneficial effects against IBD via various mechanisms, including modulation of the intestinal microflora, alteration of their metabolic features, reduction of oxidative stress, and improvement of the host’s immune responses [25,26]. Several in vitro and in vivo studies on *L. plantarum* strains have demonstrated the functional ability of LAB in IBD therapeutics by normalizing the disease activity index (DAI) score, notably suppressing the expression of pro-inflammatory factors, modulating gut microbiota, and obstructing the activation of the nuclear factor κB (NF-κB) signaling pathway [27,28,29]. Therefore, *L. plantarum* may be an effective therapy for patients with IBD. In particular, some LAB including *L. plantarum*, can produce strong antioxidant vitamins, such as riboflavin (vitamin B_2_) [30,31,32].

Riboflavin is the precursor of flavin adenine dinucleotide (FAD) and flavin mononucleotide (FMN) [33]. Both riboflavin and FAD play major roles in regulating phagocytic NADPH oxidase, which is responsible for superoxide anion radical generation in response to infection [34]. Riboflavin triggers neutrophil and macrophage proliferation in addition to phagocytosis [35]. It reduces inflammatory reactions by inhibiting infiltration and neutrophil migration, as well as the cohesion of activated granulocytes in peripheral blood [36]. Inhibition of lipopolysaccharide (LPS)-induced reactive oxygen species formation and NF-kB activation by riboflavin results in the downregulation of nitric oxide (NO) and tumor necrosis factor (TNF)-α in experimental models [37].

The objective of this study was to demonstrate alleviative effects in a DSS-induced chronic colitis murine model in BSFL extract fermented with *L. plantarum*, which increased the content of riboflavin, FAD and FMN, and explore epithelial barrier function, histopathological alterations, inflammatory cytokine levels, and tight junction (TJ) protein expression. We also aimed to determine whether the imbalanced gut microbiota composition caused by DSS-induced colitis could be recovered upon oral administration of fermented BSFL.

## 2. Materials and Methods

### 2.1. Biological Materials

BSF Hi larvae were provided by a commercial supplier in South Korea (Signalcare, Cheongdo, Republic of Korea). The larvae were reared in a controlled environment at 28–30 °C and approximately 60% humidity for 15–16 days. The larvae of BSF were fed with agroindustry by-products for 16 days and then turned in BSFL powder by drying the larvae (1 kg) in a microwave oven (P70D20TL-D4 Galanz Microwave, Foshan, China) set to 60 °C for 12 h, pulverizing (M-203 Miller, Wenling Linda Co., Ltd., Xiamen, China), and defatting (M-202 oil expeller, Wenling Linda Co., Ltd., China). The powder was stored at −20 °C until further use.

### 2.2. Riboflavin-Producing L. plantarum Isolation and Identification

To develop riboflavin-enriched fermented foods and/or feeds in the future, we screened noble lactic acid bacteria (LAB) with riboflavin-producing ability using a PCR-based method. A strain of *L. plantarum* (KCCM12757P) was isolated from the sediment collected from a fish farm tank at the National Institute of Fisheries Science (Pohang, Republic of Korea). *L. plantarum* was cultured on De Man, Rogosa, and Sharpe (MRS) broth at 30 °C for 24 h. Then, its genomic DNA was extracted using the Wizard Genomic DNA Purification kit (Promega, Madison, WI, USA). The 16s rDNA gene was amplified using 27F (5′-AGA GTT TGA TCC TGG CTC AG-3′) and 1492R (5′-TAC GGC TAC CTT GTT ACG ACT T-3′) using an AccPower PCR Premix kit (Bioneer, Daejeon, Republic of Korea) on a PCR system (Takara Bio Inc., Otsu, Japan). A homology search was performed using the Basic Local Alignment Search Tool program of NCBI (https://blast.ncbi.nlm.nih.gov/Blast.cgi, accessed on 13 January 2023). The 16s rDNA of one selected isolate showed 99% similarity with *L. plantarum* (MT597760). Expression of the riboflavin genes for riboflavin biosynthesis, including *ribA*, *ribB*, *ribC*, *ribD*, *ribF*, *ribH*, and *ribT*, were validated using PCR. Total RNA and genomic DNA of *L. plantarum* cultured in MRS were isolated using the TRIzol RNA (Invitrogen, Carlsbad, CA, USA) and Wizard gDNA purification kits (Promega), respectively, according to the manufacturer’s instructions. cDNA was synthesized using the PrimeScript 1st strand cDNA synthesis kit (TAKARA, Tokyo, Japan), and then each cDNA and gDNA was subjected to PCR using the AccPower PCR Premix kit (Bioneer, Daejeon, Republic of Korea) according to the manufacturer’s instructions. The PCR conditions were standardized as follows: initial denaturation at 95 °C for 5 min, 28 cycles of denaturation at 95 °C for 30 s, annealing at 55 °C for 30 s, extension at 72 °C for 90 s, and final extension at 72 °C for 5 min. The primer sequences for the riboflavin genes were designed based on the genomic sequence of *L. plantarum* (AP018405), which contains the riboflavin genes.

### 2.3. Quantitative Analysis of Vitamin B_2_ from L. plantarum

Hi extracted in hot water (HeHi) was used as the medium for the inoculation of *L. plantarum* (KCCM12757P), which was prepared in distilled water (5% *w*/*v*) and sterilized at 121 °C for 15 min in an autoclave (PAC-100, Suwon, Republic of Korea). Non-cultured HeHi was used as the control. The isolated *L. plantarum* strain (1 × 10^6^ CFU/mL) was inoculated into 200 mL of HeHi (1% *w*/*w* or *w*/*v*) and then incubated anaerobically at 30 °C for 24 h. After incubation, the supernatants of HeHi cells cultured with *L. plantarum* (HeHi_Lp) and non-cultured Hi (HeHi) were collected by centrifugation at 3000× *g* for 10 min. Vitamin B_2_ analysis, including FMN, riboflavin, and FAD, was performed using high-performance liquid chromatography (HPLC) based on previous studies, with some modifications [38,39,40,41]. Briefly, supernatants of HeHi and HeHi_Lp were filtered through a 0.22-µm syringe filter (Merck Millipore, Burlington, MA, USA), and reagent-grade riboflavin, FMN, and FAD (Sigma Chemical Co., St. Louis, MO, USA), dissolved in methanol (0.5% *w*/*v*), were used as reference standards. The mobile phase consisted of a mixture of methanol-10 mM NaH_2_PO_4_ (pH 5.5) solution (35/65; *v*/*v*). The filtered HeHi and HeHi_Lp supernatants were analyzed using a CapceIIPAK UG120 C18 (4.6 mm × 250 mm, 5-µm; Osaka SODA, Osaka, Japan) maintained at 40 °C during the analysis. The flow rate, injection volume, and detection wavelengths were 0.8 mL/min, 10 µL, and 445 and 530 nm, respectively. A calibration curve was constructed by plotting the peak area versus concentration, and the slope, intercept, and correlation coefficients of the calibration curve were determined. The analysis was performed in triplicate. To investigate the biological activities of HeHi_Lp, the samples were lyophilized and stored at −70 °C until use.

### 2.4. Animals

Seven-week-old male C57BL/6 mice were purchased from Orient Bio (Seongnam, Republic of Korea). All animal procedures were approved by the Animal Care and Use Committee of the Korea Food Research Institute (KFRI-M-22032). The mice were kept in a 12 h light/dark cycle under a constant temperature condition of 20 ± 3 °C with free access to reverse osmosis (RO) water and standard feed. The animals were acclimatized for one week under these conditions.

### 2.5. Murine Model of Colitis and Treatment Regimen

Chronic colitis was induced by administering 2% (*w*/*v*) dextran sodium sulfate (DSS, molecular weight 36,000–50,000 Da; MP Biochemicals, Santa Ana, CA, USA) via drinking water in a cyclical manner, i.e., three cycles with drinking periods of 1–5, 11–13, and 21–23 days. Distilled water was consumed during the remaining period (Figure 1A) [42,43,44,45]. After one week of acclimatization, the mice were randomly divided into four groups (*n* = 8/group): (1) normal control (NC) group: oral administration of vehicle only without DSS; (2) DSS-induced colitis (DSS) group: oral administration of 200 µL vehicle only; (3) HeHi_Lp_L group: treated with a low dose (1000 mg/kg of body weight) of HeHi_Lp; and (4) HeHi_Lp_H group: treated with a high dose (2000 mg/kg of body weight) of HeHi_Lp. Defatted and freeze-dried HeHi_Lp powder was dissolved in sterile drinking water. Then, the HeHi_Lp solution was given oral administration using zonde to animals.

HeHi_Lp_L (1000 mg) contained 13.26 ± 0.99, 23.99 ± 0.80, and 8.37 ± 0.42 g of riboflavin, FAD, and FMN, respectively, and HeHi_Lp_H (2000 mg) contained twice the amount of the respective components. The preventive properties of HeHi_Lp treatment were evaluated using clinicopathological profiles of body weight, colon length, and histological scoring.

### 2.6. DAI Measurement

To assess colitis severity, the DAI score was monitored daily during the experiment. DAI was scored using the method described by Friedman et al. (Table 1) [46]. The DAI was calculated based on weight loss, stool condition, and the presence or absence of blood; scores were assigned according to each condition and then summed. In the case of weight loss, 0% scored 0 points, 0–10% scored 1 point, 11–15% scored 2 points, 16–20% scored 3 points, and >20% scored 4 points. For the stool condition, 0 points were assigned for normal cases, 2 points for slightly watery conditions, and 4 points for diarrhea. The absence and presence of hematochezia merited 0 and 4 points, respectively. Weight was measured before the experiment to minimize the error of weight by group, and weight was measured at the same time every day during the experiment.

### 2.7. Blood Collection and Organ Sampling

Blood was collected from the inferior vena cava of mice under anesthesia and placed in serum-separating tubes. The collected blood was kept at 25 °C for approximately 30 min and then centrifuged at 3000 rpm for 15 min. The serum was collected and stored in an ultralow-temperature freezer at −80 °C. Immediately after blood collection, the spleen was extracted and weighed. The colon was separated, and its length was measured (from the cecum to the anus) after immersing it in saline to remove blood and fat. The moisture was removed using moisture absorption paper. The cecum and anus were also separated, and after separation, the inside of the large intestine was washed with saline, the proximal colon (half the upper part of the entire length) was stored in a deep freezer at −80 °C, and the distal colon (half the lower part of the entire length) was preserved using a 10% formalin solution.

### 2.8. Histopathological Analysis

The tissues of distal colon sections were washed with BPS and fixed in 10% formaldehyde, and then embedded in paraffin blocks. Paraffin-embedded sections of distal colon tissue were stained with hematoxylin and eosin (H&E) for histopathological analyses. Using an optical microscope (Eclipse 80i, Nikon Inc., Melville, NY, USA; ×40 magnification), sections were evaluated for histological changes in different observation fields randomly selected according to the severity of colitis.

### 2.9. Western Blotting

RIPA (radioimmunoprecipitation assay) lysis buffer (Rockland, Inc., Gilbertsville, PA, USA) containing phenylmethanesulfonyl was used to extract the protein of colons. The protein concentration was assessed using the Bio-Rad Protein Assay (Bio-Rad Laboratories, Inc., Hercules, CA, USA). Similarly, 30 μg of tissue lysates was loaded onto 12% SDS-PAGE gels, separated, and transferred to a PVDF membrane. The membrane was blocked with blocking solution (Bio-Rad Laboratories, Inc., Hercules, CA, USA) for 1 h at room temperature, then incubated with 1:1000-diluted primary antibodies: iNOS (Cell Signaling Technology, Inc., Beverly, MA, USA), COX-2 (Cell Signaling Technology, Inc., Beverly, MA, USA), and β-actin (Cell Signaling Technology, Inc., Beverly, MA, USA) at 4 °C overnight for 18 h. Following washing with TBST, the membrane was then incubated with 1:10,000-diluted goat anti-rabbit IgG-HRP-conjugated secondary antibody (Cell Signaling Technology, Inc., Beverly, MA, USA) for 1 h at room temperature. The bands were detected using an enhanced chemiluminescence detection kit—Clarity Max western ECL substrate (Bio-Rad Laboratories, Inc., Hercules, CA, USA)—and were analyzed using ImageJ software (Version 1.52; National Institutes of Health, Bethesda, MA, USA).

### 2.10. Real-Time PCR

Total RNA was isolated using a PureLink RNA Mini Kit (Invitrogen, Carlsbad, CA, USA). One milliliter of RNAiso Plus (Takara, Otsu, Japan) was added to 0.1 g frozen colon tissue. The tissue was homogenized and allowed to settle at 25 °C for 5 min. Chloroform (200 µL) was added to the tissue, mixed vigorously, allowed to settle at 25 °C for 5 min, and centrifuged at 10,000 rpm at 4 °C for 10 min. The separated supernatant was collected, and an equal volume of 70% ethanol was added followed by centrifugation at 10,000 rpm at 4 °C for 10 min to obtain an RNA pellet. cDNA was synthesized using the RNA Maxime RT Premix Kit (iNtRON Biotechnology, Seongnam, Republic of Korea). A mixture of 1 µL of Oligo dT Primer (50 µM), 10 µM of dNTP Mixing, and 1 µg of template RNA was prepared. The volume adjusted to 20 µL with RNase free dH_2_O and reacted at 45 °C for 60 min to synthesize cDNA; enzyme activity was inhibited for 5 min at 95 °C. The synthesized cDNA was stored at −20 °C and used as a template when performing real-time PCR. The cDNA synthesized with the template was analyzed for mRNA expression using the Power SYBR™ Green PCR Master Mix (Applied Biosystems, Waltham, MA, USA). The primers used to analyze the expression of each gene were synthesized by Macrogen (Seoul, Republic of Korea). After initial denaturation at 95 °C for 10 min, the reaction was performed 40 times for 15 s at 95 °C and 1 min at 60 °C as one cycle. The values of the threshold cycle and C(t) represented by sensing the fluorescence signal for each cycle were analyzed, and the expression of mRNA in each experimental group were quantitatively analyzed using the CFX96 real-time system (Bio-Rad, Hercules, CA, USA). β-actin and GAPDH were used as the internal transcription markers; the gene primer base sequences used during amplification are shown in Table 2.

### 2.11. Next-Generation Sequencing (NGS) Analysis

Fecal samples for microbiota analysis were collected for each group using sterile forceps the day before dissection and stored at −80 °C until further use. Here, we used one pooled fecal sample of each group for microbiota analysis, since it was difficult to obtain enough material in mice with colitis. Total DNA from the collected fecal samples was extracted using a DNeasy PowerSoil Kit (Qiagen, Hilden, Germany) according to the manufacturer’s instructions and quantified using Quant-IT PicoGreen (Invitrogen). The sequencing libraries were prepared based on the Illumina 16S Metagenomic Sequencing Library protocols to amplify the V3 and V4 regions (V3-F:5′-TCGTCGGCAGCGTCAGATGTGTATAAGAGACAGCCTACGGGNGGCWGCAG-3′ and V4-R:5′-GTCTCGTGGGCTCGGAGATGTGTATAAGAGACAGGACTACHVGGGTATCTAATCC-3′). Paired-end (2 × 300 bp) sequencing was performed by Macrogen using a MiSeq platform (Illumina, San Diego, CA, USA). After removing the sequencing adapter sequence and the F/R primer sequence of the target gene region using the Cutadapt (v3.2) program [47], the forward sequence (Read1) and reverse sequence (Read2) were cut to 250 bp and 200 bp, respectively. To compare and analyze the microbial communities, subsampling was applied and normalized based on the minimum number of reads of samples using the QIIME (v1.9) program [48]. Various microbial communities were compared and analyzed using QIIME with amplicon sequence variant abundance and taxonomic information.

### 2.12. Statistical Analysis

Statistical analyses were performed using GraphPad Prism 8.0. Data are presented as means (SEM). Differences among groups were compared using one-way analysis of variance (ANOVA). One-way ANOVA was used for normally distributed data. Statistical significance was set at *p* < 0.05.

## 3. Results

### 3.1. Vitamin B_2_ Production by L. plantarum

Through in silico analysis based on the genomic sequences of LAB, approximately 95 LAB collected from the sediment of a fish farm tank were selected as raw materials for screening riboflavin-producing LAB. LAB 16s rDNA sequencing identified 56 *Lactobacillus sakei*, 7 *L. brevis*, 6 *L. plantarum*, 1 *Lactococcus lactis*, and 12 *Leuconostoc*. Some LAB contain riboflavin (rib) biosynthesis genes, including *ribA*, *ribB*, *ribC*, *ribD*, *ribF*, *ribH*, and *ribT* [38,49]. The rib genes of *L. plantarum* (KCCM12757P) were identified and their expression was analyzed in both the gDNA and cDNA of the organism. *ribA*, *ribB*, *ribC*, *ribD*, *ribF*, *ribH*, and *ribT* showed remarkable expression in gDNA; only *ribA*, *ribC*, and *ribH* were expressed in cDNA. Vitamin B_2_ production from riboflavin, FAD, and FMN produced by fermentation for 24 h by *L. plantarum* was analyzed in HeHi. The riboflavin, FAD, and FMN levels of HeHi as determined using HPLC were 7.98 ± 0.42, 10.64 ± 0.66, and 2.66 ± 0.32 µg/g respectively, and levels in HeHi_Lp were 13.26 ± 0.99, 23.99 ± 0.80, and 8.37 ± 0.42 µg/g, respectively (Table 3). The riboflavin, FAD, and FMN contents of HeHi_Lp fermented by *L. plantarum* were approximately 1.7-, 2.3-, and 3.1-fold higher, respectively, then those of HeHi. Fermented HeHi_Lp showed a 2.1-fold increase in vitamin B_2_ content. HeHi and HeHi_Lp were used in subsequent experiments.

### 3.2. Effect of HeHi_Lp on DSS-Induced Colitis Symptoms

First, we explored the preventive effects of HeHi_Lp on body weight and DAI in DSS-induced murine colitis. Treatment with 2% DSS for three cycles resulted in three main colitis symptoms—significant weight loss, diarrhea, and hematochezia—whereas normal mice did not show these symptoms (Figure 1B–E). However, DSS-treated mice treated with HeHi_Lp (L and H) showed less weight loss than mice treated with DSS alone. Moreover, body weight restoration was remarkably more enhanced in HeHi_Lp_H-treated mice than in DSS-treated mice. On day 29, the weight change was highest at 113.8 ± 4.0% in NC, and in the DSS group, it was significantly reduced to 94.1 ± 10.7% (*p* < 0.001, Figure 1B,C). In the HeHi_Lp_L group, body weight tended to increase to 102.1 ± 2.2%, but no significant difference was confirmed; in the HeHi_Lp_H group, body weight significantly increased to 109.7 ± 4.8% (*p* < 0.01).

The DAI calculated in this experiment showed a trend conforming to the results of the weight change, i.e., DAI increased with weight loss and decreased with weight gain (Figure 1D,E). In all the treated groups, DAI increased and then decreased every three cycles. Checking each DAI score on day 28, the day before euthanasia, a significant difference was observed: the score was 0 points for the NC group and 3.4 ± 1.5 points for the DSS group (*p* < 0.01), 0.8 ± 1.3 points (*p* < 0.05) for the HeHi_Lp_L group, and 0.4 ± 0.9 points (*p* < 0.01) for the HeHi_Lp_H group.

### 3.3. Effect of HeHi_Lp on Colon Histopathological Alterations and Spleen Weight

Since the change in bowel length in the colitis animal model is related to the progression of inflammation, this study confirmed the effect of the administered treatments on changes in colon length, spleen weight, and mucosal integrity in the DSS-administered animal model. The large intestine length in the DSS group (5.84 ± 0.30 cm) was significantly shorter than in the NC group (8.08 ± 0.33 cm, *p* < 0.001, Figure 2A,B); the lengths were 6.02 ± 0.18 cm (*p* < 0.05) and 6.78 ± 0.29 cm (*p* < 0.01) in the HeHi_Lp_L and HeHi_Lp_H groups, respectively.

The spleen weight/body weight of mice with DSS-induced colitis was significantly higher (0.0102 ± 0.0038, *p* < 0.001) than that of normal mice (0.0026 ± 0.0002) and was significantly lower in the HeHi_Lp_L (0.0070 ± 0.0017, *p* < 0.05) and HeHi_Lp_H groups (0.0061 ± 0.0012, *p* < 0.01) (Figure 2C).

Histological examination of the tissue samples revealed no abnormalities in colon tissue of the NC group. In contrast, destruction of the nucleus, infiltration of inflammatory cells in mucous membranes, and loss of epithelial tissue were observed in the DSS group. The tissue in the HeHi_Lp_L and HeHi_Lp_H groups was found to be partially damaged compared to the NC group, but DSS-induced damage was found to be recovered (Figure 2D).

### 3.4. HeHi_Lp Administration and Inflammatory Mediators in Colon Tissue

Western blotting analysis revealed that the protein expression of cyclooxygenase 2 (COX-2) and inducible nitric oxide synthase (iNOS), which are representative inflammatory indicator factors, in the DSS group increased significantly compared to that in the NC group (Figure 3A,B, *p* < 0.001). In the HeHi_Lp groups, the expression levels of COX-2 and iNOS decreased significantly (*p* < 0.001).

In order to confirm whether HeHi_Lp administration also affects the gene expression of inflammatory cytokines, the mRNA expression of TNF-α, IL-6, and IL-1 was measured in the colon tissue of mice with DSS-induced colitis using real-time PCR. TNF-α, IL-6, and IL-1β mRNA expression significantly increased in the DSS group compared with that in the NC group (Figure 3C–E). Compared with that in the DSS group, the gene expression of TNF-α, IL-6, and IL-1β decreased in the HeHi_Lp_L and HeHi_Lp_H groups (Figure 3C–E); however, the decrease in the expression level of TNF-α was not significant in the HeHi_Lp_L group (Figure 3C). The gene expression levels of IL-6 and IL-1β in the colon tissue significantly reduced in the HeHi_Lp_L and HeHi_Lp_H groups. IL-6 in the HeHi_Lp_L group and both IL-6 and IL-1β in HeHi_Lp_L and HeHi_Lp_H groups were comparable to those in the NC group (Figure 3D,E).

### 3.5. HeHi_Lp Supplementation Effect on Intestinal Barrier Function

The epithelial cell layer is interconnected by TJ proteins, such as zonula occludens 1 (ZO-1), occludins, and claudins, which are crucial for the maintenance of epithelial barrier integrity. Gene expression of ZO-1, occludin, and claudin 1, which are closely linked proteins, in the intestinal tissue was confirmed by real-time PCR to confirm the integrity of the mucosal barrier during colon inflammation induced by DSS (Figure 4A–C). PCR experiments showed that the remarkably decreased expression of ZO-1, occludin, and claudin 1 genes by DSS were significantly increased in the HeHi_Lp_H group.

### 3.6. Effect of HeHi_Lp Treatment on Gut Microbiota Composition

NGS was used to examine the effects of HeHi_Lp on gut microbiota. By analyzing the bacterial distribution in the fecal samples of each group, the relative distribution ratio of bacteria occupying the microbiological classification stage at the family (Figure 5A) and species levels (Figure 5B) was determined. In the DSS group, the relative abundance of several beneficial bacteria such as *Lachnospiraceae*, *Muribaculum intestinale (Muribaculaceae* family), *Lactobacillaceae* and *Bifidobacteriaceae* was lower than that in the NC group, confirming DSS induction. The relative abundance of *Muribaculaceae* and *Lactobacillaceae* was restored after supplementation with HeHi_Lp compared with their abundance in the DSS group, suggesting favorable efficacy of HeHi_Lp. Notably, the relative abundance of *L. johnsonii (Lactobacillaceae* family) in the DSS group increased after administration of HeHi_Lp. However, the relative abundance of bacteria associated with intestinal inflammation such as *Bacteroides caccae* (*Bacteroidaceae* family), *Clostridiaceae*, and *Akkermansia muciniphila* (*Akkermansiaceae* family) increased in the DSS group compared with the NC group. HeHi_Lp administration restored the disruption in the abundance of all the aforementioned bacteria.

## 4. Discussion

The pathogenesis of IBD is complex and involves environmental, microbial, genetic, and immune factors, and its prevalence has increased not only in humans but also in pets [2,50]. IBD decreases the quality of life due to a low remission rate and may lead to the development of colorectal cancer [51]. Moreover, the WHO has reported that it has a high relapse rate and currently has no effective remedy [52]. Therefore, effective treatment of IBD is essential to improve the quality of life and prevent the development of colorectal cancer.

In this study, we identified the long-term preventive function of fermented *H. illucens* larvae with *L. plantarum* KCCM12757P (HeHi_Lp) in enhancing clinical effects in a DSS-induced chronic colitis murine model. The colitis model induced by DSS showed clinical symptoms characterized by loss of body weight, diarrhea, and hematochezia in mice. The effects of HeHi_Lp were examined, and each symptom was converted to a calculable score. Biochemical and physiological processes in chronic colitis have been measured using a DSS-induced chronic colitis mouse model as it provides a comprehensive overview of prospective therapeutic approaches [4,53,54]. Body weight changes and the presence of diarrhea and bloody stools were used to measure the DAI. Damage and inflammatory reactions of colonic tissue were assessed by H&E staining. Loss of body weight, shortened colon length, diarrhea, and increased DAI scores have been previously reported in DSS-administered mice [23,55].

Damaging effects, such as a marked decrease in body weight and increased DAI, were observed during cycle 2. At both concentrations of HeHi_Lp administered during the three-cycle period, the clinical symptoms tended to recover, and in the high-concentration group, both weight change and DAI measurements were statistically significant. Moreover, the damage to the colon tissue observed in the DSS group was significantly recovered in both the HeHi_Lp (L and H) groups. Further, the anti-inflammatory effect was confirmed by reduced spleen/body weight ratio after HeHi_Lp administration at both concentrations. These results are consistent with those of previous findings [56,57]. Therefore, it can be concluded that HeHi_Lp supplementation has a preventive effect against the clinical symptoms of DSS-induced chronic colitis.

Under abnormal conditions, increased secretion of pro-inflammatory cytokines peaks during an inflammatory response [57,58,59,60]. COX-2 and iNOS are inducible enzymes that are predominantly expressed at the sites of inflammatory response; COX-2 and iNOS activation produces excessive inflammatory mediators that may contribute to intestinal damage. Additionally, iNOS acts in synergy with COX-2 to promote inflammation [61,62]. TNF-α and IL-6, pleiotropic cytokines produced by activated mononuclear cells, participate in IBD-associated inflammation; their excessive production has been linked to IBD pathogenesis [63]. It is well known that p38 mitogen-activated protein kinase (MAPK) is a key factor in the regulation of pro-inflammatory cytokines, including TNF-α and IL-6, in the mucosal tissue devastation in IBD patients [64].

After the induction of chronic colitis, a substantial elevation in the levels of iNOS, COX-2, TNF-α, IL-6, and IL-1β were observed in this study. However, HeHi_Lp effectively decreased their levels, which is in agreement with the results of a previous study [59]. In particular, at high concentrations of HeHi_Lp (HeHi_Lp_H), statistically significant differences were confirmed for all indicators compared with the DSS group. Therefore, HeHi_Lp supplementation may prevent DSS-induced chronic colitis by regulating cytokine levels.

The gut epithelium plays a vital role in IBD pathogenesis. Intestinal permeability is mainly controlled by TJs, which regulate epithelial barrier function [57]. The representative features of IBD are abnormal intestinal epithelial TJs, which are characterized by low expression of intestinal epithelial TJ-related proteins [65,66]. Peripheral membrane proteins (ZO-1), transmembrane proteins (occludin and claudins), and intracellular regulatory molecules are representative tight-ion proteins [67] and TJ damage can occur at the outbreak of colitis. ZO-1 and occludin play multiple roles in intestinal homeostasis by regulating cell bypass barrier function and optimizing the action of TJ protein molecules [68]. Claudins are other essential inflammatory regulators that may have anti-inflammatory activity [69]. In our study, the colonic expression of the three TJ proteins, ZO-1, occludin, and claudin 1, was reduced in the DSS-treated groups. However, co-treatment of mice with DSS and HeHi_Lp_H caused a marked increase in these protein levels in the colon tissue, suggesting their potential role in improving intestinal stability. The increased mRNA expression of these proteins after administration of HeHi_Lp_H is consistent with the observations of previous studies [56,70]. Taken together, our biochemical analyses results suggested that HeHi_Lp regulated the inflammatory mediators (COX-2 and iNOS), cytokines (TNF-α, IL-6, and IL-1β) and TJ-related proteins (ZO-1, occludin, and claudin-1) to maintain intestinal mucosal barrier function, and eventually alleviated histopathological damages in colon by DSS induction.

Intestinal integrity is also closely related to the status of the intestinal immune system, which is significantly controlled by gut microbial distribution and homeostasis. The microbial community of a healthy gut plays an essential role in protecting the host against invasion by pathogenic microorganisms and regulating beneficial intestinal metabolites [71,72]. We explored the distribution of the microbial community in all studied groups at the family and species levels (Figure 5A,B), and these distributions were found to vary in all studied groups. As expected, in the NC group, the relative abundance of several beneficial bacteria such as *Lachnospiraceae*, *M. intestinale*, *Lactobacillaceae*, and *Bifidobacteriaceae* reduced, but some bacteria, such as *B. caccae*, *Clostridiaceae*, *Oscillospiraceae*, and *A. muciniphila*, related to intestinal inflammation, increased compared with the DSS group, confirming colitis induction. In particular, the relative abundance of *Lachnospiraceae*, which is associated with the metabolism of butyrate and propionate [73], the major short-chain fatty acids (SCFAs), was lower in the DSS group (8.5%) than the NC group (9.3%). *M. intestinale* also produces SCFAs [74], and its relative abundance was lower in the NC group (0.7%) than the DSS group (11.1%) (16.1-fold difference). In particular, the relative abundance of *M. intestinale* in DSS group, which had decreased, was restored to 8.1% and 5.9% after administration of HeHi_Lp_L and HeHi_Lp_H, respectively.

SCFAs are involved in ameliorating lipid metabolism, regulating energy consumption, maintaining the integrity of the intestinal mucosa, and regulating the intestinal pH, the immune system, and the inflammatory response [74]. Previous studies have described possible IBD treatment with colonic gut microbiota via their SCFA production [75]. The intestinal SCFAs via gut microbiota are involved in intestinal and immune homeostasis [76] and protect the intestinal mucous membrane and TJs. Furthermore, SCFAs relieve colitis by inhibiting the NF-κB signaling pathway, which is closely associated with inflammatory cytokines and chemokines [77,78,79]. In particular, *L. plantarum*, due to its ability to synthesize high levels of conjugated linoleic acids, could alleviate colitis by hindering the NF-κB signaling pathway [80]. Furthermore, the relative abundance of *Lactobacillaceae and Bifidobacteriaceae*, which are representative probiotics [81,82], decreased from 24.6% to 21.8% and from 6.1% to 0.2% in the NC and DSS groups, respectively. The relative abundance of *Lactobacillaceae* in the DSS group recovered to 55.8% and 41.8% after administration of the HeHi_Lp_L and HeHi_Lp_H, respectively. However, colitic mice of the DSS group showed 4.8-, 2.1-, and 1.7-fold higher abundance of bacteria related to intestinal inflammation such as *B. caccae*, *Clostridiaceae*, and *A. muciniphila* than the non-colitic control mice of the NC group, respectively. Interestingly, the upregulated abundance of all bacteria reduced markedly in the HeHi_Lp_L and H groups.

*B. caccae* is known to cause IBD by overproducing succinic acid [83], and *Clostridiaceae and A. muciniphila (Akkermansiaceae* family) are well known as colitogenic and dysbiotic bacteria [84,85]. Consistent with our results, *Akkermansia* shows a positive correlation with IBD patients, which may due to the mucin-degrading ability of *Akkermansia* and its highly immunostimulatory LPS activity [86], resulting in a high production of pro-inflammatory cytokines. Mucin is the major component of colonic mucosa, and the increased abundance of *Akkermansia* may contribute to mucosal damage [87]. In this study, the administration of HeHi_Lp decreased both pro-inflammatory cytokines and the abundance of *Akkermansia.*

These results indicate that HeHi_Lp exerts a protective effect on DSS-induced chronic colitis by modulating the gut microbiota structure to improve SCFA levels, which could reduce inflammation and enhance intestinal barrier functions. On the other hand, there was difficulty in the process of sampling the fecal samples used in this gut microbiota composition analysis. The reasons are that the consistency and amount of stool were significantly decreased by chronic inflammation. To solve these problems, all animals in each group were raised in one metabolic cage to induce the same environmental conditions to be maintained by exposing them to feces and urine, etc., and the minimum amount of feces that could be analyzed was collected and used for analysis. Therefore, the results of this experiment should be comprehended as being of a preliminary nature, as a single sample for each group may lack statistical significance.

HeHi_Lp is a natural material increased riboflavin, and FMN and FAD through fermentation of HeHi with *L. plantarum* KCCM12757P in an optimal process. FMN plays a major role as an enzyme cofactor along with FAD, another molecule originating from riboflavin [33]. Riboflavin is converted into two active coenzymes in the body, FMN and FAD, and these coenzymes are involved in energy metabolism and numerous enzyme reactions. FMN and FAD function as strong antioxidants and help neutralize harmful free radicals in the body [88]. They work with other antioxidants to protect cells from oxidative damage. Therefore, FMN and FAD are absorbed into the body according to the intake of HeHi_Lp, protecting colon tissue and inducing normalization of intestinal microorganisms by suppressing inflammation caused by DSS.

## 5. Conclusions

In this study, we demonstrated that fermented *H. illucens* larvae with *L. plantarum* KCCM12757P (HeHi_Lp) prevented DSS-induced colitis by reducing colitis symptoms and pro-inflammatory cytokine levels and upregulating the mRNA expression levels of TJ proteins. HeHi_Lp administration also modulated gut microbial distribution. Therefore, the protective functions of HeHi_Lp may have novel applications in the pet-food industry.

## Figures and Tables

**Figure 1 antioxidants-12-01822-f001:**
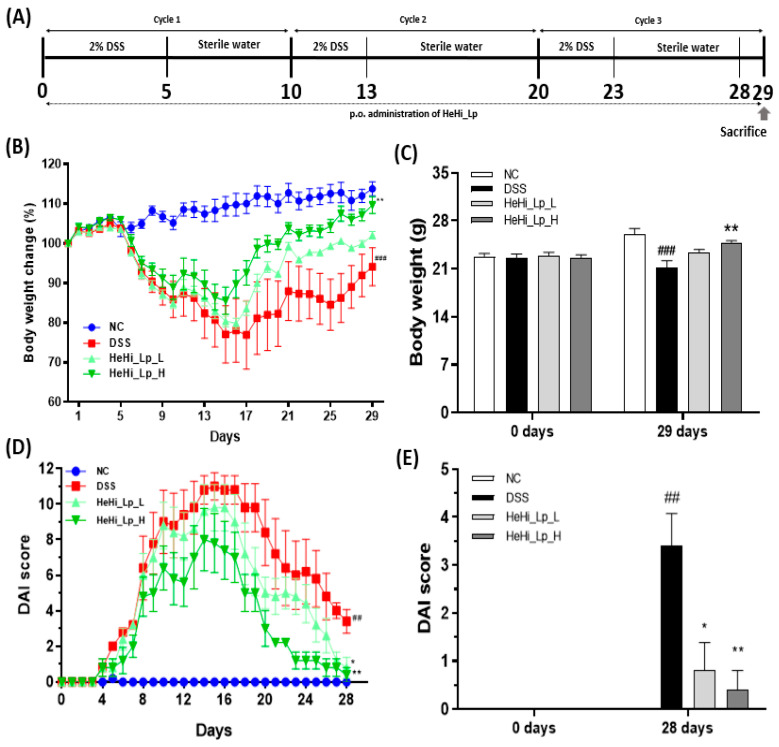
The effect of HeHi_Lp administration on symptoms in mice with DSS-induced chronic colitis. A schematic diagram of the overall experiment. The time frame for the cycle, treatment period, and animal euthanasia is described (**A**). Effect of HeHi_Lp on DSS-induced changes in body weight of mice (**B**,**C**). Effect of HeHi_Lp on the disease activity of DSS-induced mice (**D**,**E**). Data are presented as means ± standard deviation (SD). ^##^
*p* < 0.01 and ^###^
*p* < 0.001 compared with the normal control group (NC); * *p* < 0.05 and ** *p* < 0.01 compared with the DSS group.

**Figure 2 antioxidants-12-01822-f002:**
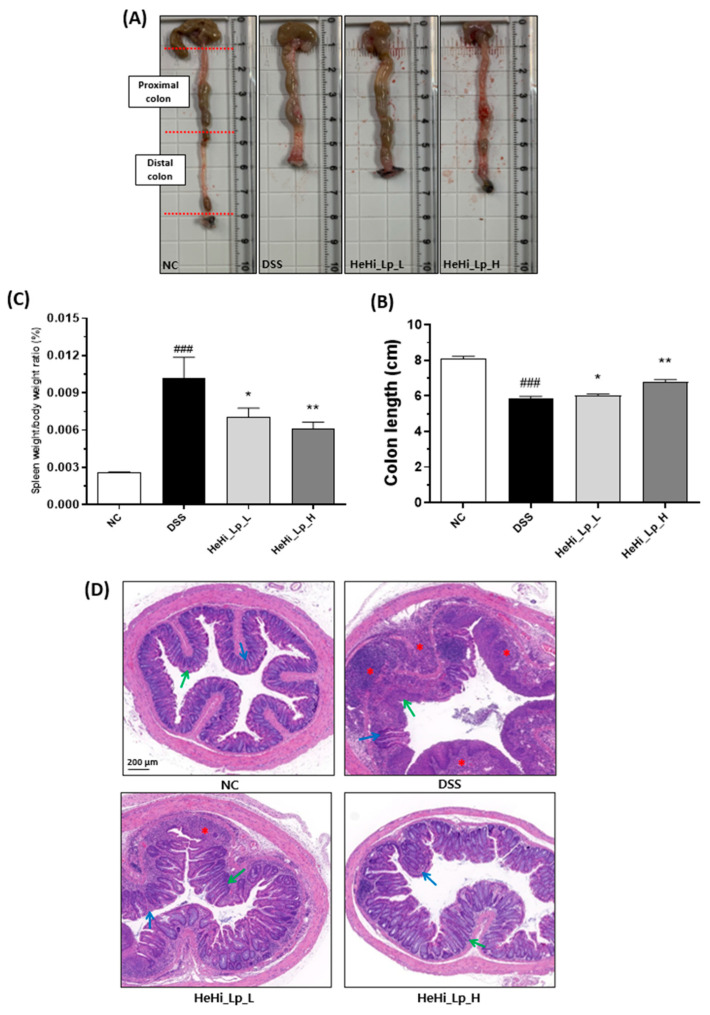
Protection effect of HeHi_Lp to DSS-induced chronic colitic mouse model. Image showing the determined colonic length of mice from each group after euthanasia (**A**). Effect of HeHi_Lp on the colonic length (**B**) and spleen weight/body weight ratio (**C**) of DSS-induced mice. (**D**) Protective effect of HeHi_Lp to histological damage and inflammation of colon in DSS-induced mice demonstrated by hematoxylin and eosin (H&E) staining (scale bar = 200 µm). Red asterisks referred to inflammatory cell infiltration. Green arrows pointed to columnar epithelial cells while blue arrows referred to crypts. Data are presented as means ± SD. ^###^
*p* < 0.001 compared with the NC group. * *p* < 0.05 and ** *p* < 0.01 compared with the DSS group.

**Figure 3 antioxidants-12-01822-f003:**
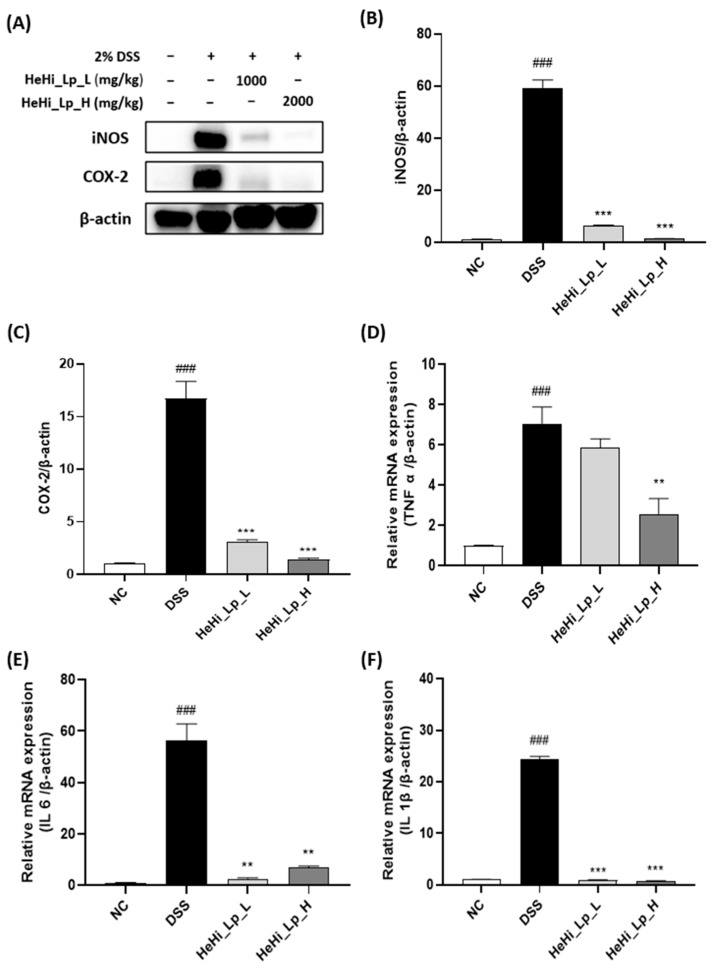
Effects of HeHi_Lp supplementation on inflammatory indicator factors and inflammatory cytokine concentrations. (**A**) Representative Western blot image, (**B**) iNOS and (**C**) COX-2 protein expression levels were determined in colon tissue by Western blot analysis. (**D**) TNF-α, (**E**) IL-6, and (**F**) IL-1β expression levels in colon tissue were detected by qRT-PCR. Data are presented as means ± SD. ^###^
*p* < 0.001 compared with the NC group. ** *p* < 0.01 and *** *p* < 0.001 compared to the DSS group. iNOS: inducible nitric oxide synthase; COX-2: cyclooxygenase 2; TNF: tumor necrosis factor; IL: interleukin.

**Figure 4 antioxidants-12-01822-f004:**
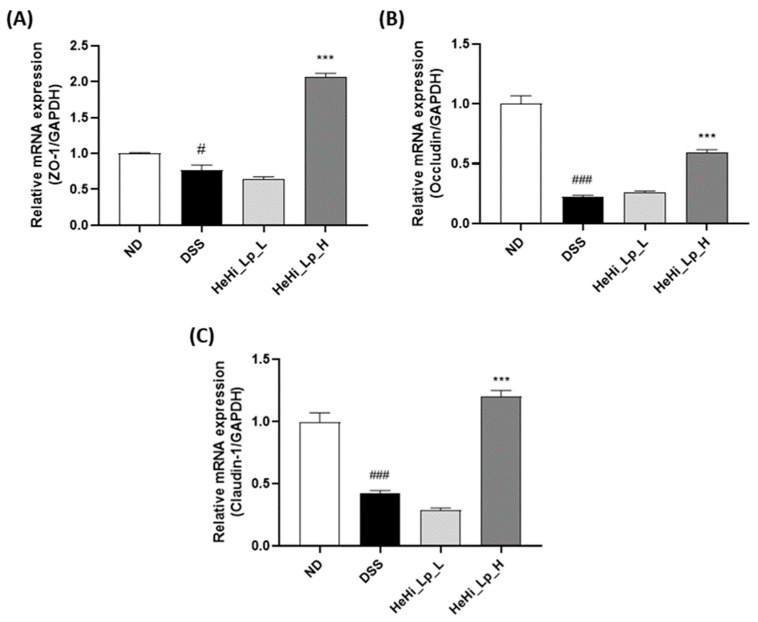
Effect of HeHi_Lp on gene expression in the colon tissue of mice with DSS-induced colitis. (**A**) Zonula occludens 1 (ZO-1), (**B**) occludin, and (**C**) claudin 1. Data are presented as means ± SD. ^#^
*p* < 0.05 and ^###^
*p* < 0.001 compared with the NC group. *** *p* < 0.001 compared to the DSS group.

**Figure 5 antioxidants-12-01822-f005:**
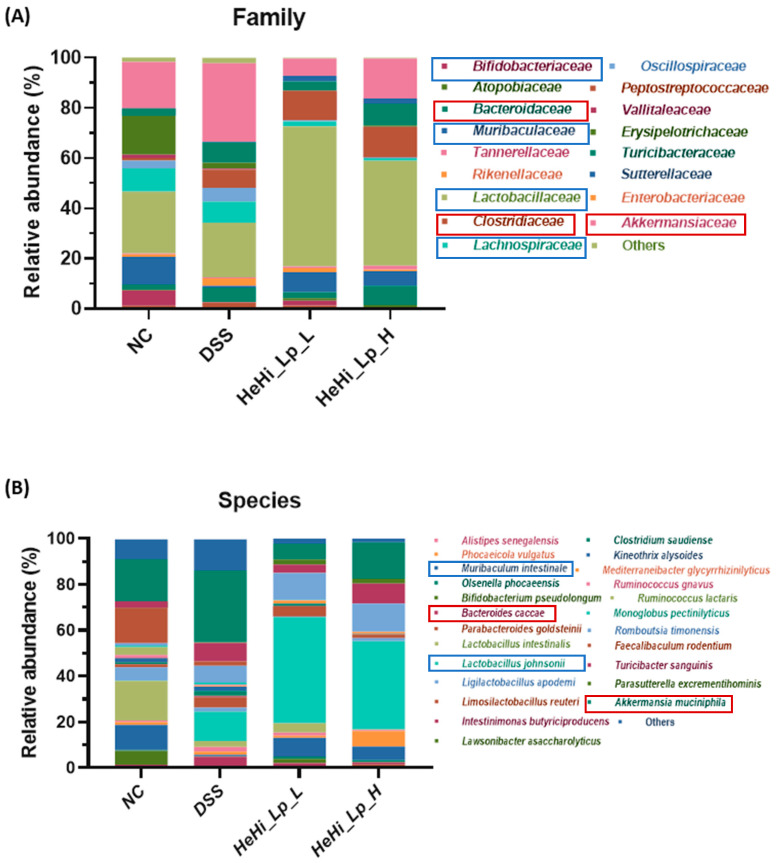
Pyrosequencing of gut microbiota. Relative sequence abundance of (**A**) bacterial families and (**B**) species. The blue box indicates beneficial bacteria, and the red box indicates bacteria associated with intestinal inflammation.

**Table 1 antioxidants-12-01822-t001:** Scoring system of the disease activity index (DAI) ^a^.

Score	Weight Loss (%)	Stool Consistency	Hematochezia ^b^
0	None	Normal	Absence
1	0–10	-	-
2	11–15	Loose stool ^c^	
3	16–20	-	
4	>20	Diarrhea	Presence

^a^ DAI = (score of weight loss) + (score of stool consistency) + (score of hematochezia). ^b^ Presence of gross blood in the stool or anus. ^c^ Formation of a stool that readily becomes a paste on the anus of mice.

**Table 2 antioxidants-12-01822-t002:** Primer sequences used to detect mRNA specific to target genes.

Gene	Sequence (5′→3′)	Accession Number
*β-actin*	F: 5′-CAGCTGAGAGGGAAATCGTG-3′ R: 5′-CGTTGCCAATAGTGATGACC-3′	NM_031144.3
*TNF-α*	F: 5′-ACCCTCACACTCAGATCATC-3′ R: 5′-GAGTAGACAAGGTACAACCC-3′	NM_012675.3
IL-6	F: 5′-TGGAGTACCATAGCTACCTG-3′ R: 5′-TGACTCCAGCTTATCTGTTA-3′	NM_012589.2
IL-1β	F: 5′-TGTAATGAAAGACGGCACAC-3′ R: 5′-TCTTCTTTGGGTATTGCTTG-3′	NM_031512.2
*ZO-1*	F: 5′-GCTTTAGCGAACAGAAGGAGC-3′ R: 5′-TTCATTTTTCCGAGACTTCACCA-3′	NM_003257
*Occludin*	F: 5′-TTGAAAGTCCACCTCCTTACAGA-3′ R: 5′-CCGGATAAAAAGAGTACGCTGG-3′	NM_002538
*Claudin-1*	F: 5′-CCCTTCAGCAGAGCAAGGTT-3′ R: 5′-TAGGGCAACCAAGTGCCTTT-3′	NM_021101

Abbreviations: F, forward primer; IL-6, interleukin 6; R, reverse primer; TNF-α, tumor necrosis factor α; ZO-1, zonula occludens 1.

**Table 3 antioxidants-12-01822-t003:** Concentrations of riboflavin produced by *Lactobacillus plantarum* in *Hermetia illucen*.

Vitamin B_2_ (µg/g)	HeHi	HeHi_Lp
Riboflavin	7.98 ± 0.42	13.26 ± 0.99
FAD	10.64 ± 0.66	23.99 ± 0.80
FMN	2.66 ± 0.32	8.37 ± 0.42
Total (µg/g)	21.28 ± 0.47	45.62 ± 0.73

HeHi, *H. illucens* extracted within hot water (5%, *v*/*v*); HeHi_Lp, HeHi fermented with *L. plantarum*. FAD, flavin adenine dinucleotide; FMN, flavin mononucleotide.

## Data Availability

Data is contained within the article.

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
