# Peer review of "Hermetia illucens Fermented with Lactobacillus plantarum KCCM12757P Alleviates Dextran Sodium Sulfate-Induced Colitis in Mice"

_antioxidants, 2023, doi:10.3390/antiox12101822_

Round 1

Reviewer 1 Report

The authors seem to be emphasizing the significance of riboflavin in treating colitis. Why was riboflavin chosen exclusively for quantification? Were other components in the fermented supernatant also measured by the authors?

In case the authors assume that riboflavin is the primary component responsible for treating colitis, it might diminish the novelty of the study, as previous research has already investigated the beneficial effects of riboflavin in colitis patients.

To comprehensively study the role of extracts fermented with Lactobacillus plantarum, it is essential to include a control group without Lactobacillus plantarum as well.

Regarding section 2.2, the authors should clarify why they opted to isolate the strain instead of using a commercially available one.

How was Heli_LP administered to the animals and in what form?

The term "protects" in the title seems inaccurate, given that the treatment follows DSS-induced colitis.

When measuring the kidney, it is crucial to account for body weight changes, but including the absolute weight of the kidney may provide more meaningful insights.

The Western images in Figure 3 appear to be missing. Additionally, using concentration for gene expression is not appropriate; protein concentration would be more suitable since structural changes are executed through proteins (Figure 4).

In Figure 5, it is suggested to present the specific good and bad bacteria separately for the readers' convenience. It would be helpful to know the sample size for sequencing.

Line 31 requires one more dot before [1]. Line 163 contains a wrong description of the DSS group. Line 309 is inappropriate to use clinical symptoms in mice, and the full name of DSS should not be repeated. Lines 370-379 should have the gene names italicized. DAI should use its full name for the first time.

Author Response

We used yellow (to reviewer 1's comments) and green (to reviewer 2's comments) highlights to mark in the revised manuscript.

Reviewer 2 Report

Comments to the Authors of manuscript number antioxidants-2535149 entitled “Hermetia illucens Fermented with Lactobacillus plantarum KCCM12757P Protects Mice Against Dextran Sodium Sulfate-Induced Colitis”.

It is a very interesting and performed well study.

1. L22- weight of what?

2.L31-dot and space

3.L 51- the examples are needed

4. L 83- it is not a real reason to perform any study- the lack of studies. The proper hypothesis is needed

5. L 84-85- the study was performed to explain…or the goal of the study was… not the paper …

6. The abstract and introduction are written well

7. L 157- DSS?

8. L 164 – 1,000 mg/kg of what? It is unclear

9. L 202-206 – there is no explanation how the colon was weighed. The abstract informs that it was done.

10. L 201- how long is colon in mice that there is spoken about distal colon?

11. The presence of Figure 1A is very helpful

12. Figure 2A – distal colon should be marked

Author Response

(The authors gave the same response as above.)

Reviewer 3 Report

The paper of Son et al is focussed on the IBD in animals and a possible treatment.The experiments were appropriate but the redaction has a lot of problems.

So, the figure 2(D) the images must be bigger and arrows highlighting the alterations must be introduced. Also, in figure 3, results of Western blot analyses are presented together with those of qPCR, but this is not specified in the figure caption.

In discussion, the author must explain clearly the role of FAD and FMN in the amelioration of IBD in correlation with the decrease of inflammation. Also at page 14, lines 225-229 have to be written because the information is unclear.

Furthermore, the histopathological amelioration has to be correlated with the biochemical analyses.

Author Response

We used green (to reviewer 1's comments) and yellow (to reviewer 3's comments) highlights to mark in the revised manuscript.

Round 2

Reviewer 1 Report

Thanks for the responses.

The critical comments were not fully addressed. Especially, the sample number for microbiota sequencing is only 1, which is not enough to draw any reliable results. Without this part, the novelty of the paper is largely compromised. 

The "metagenomics" is misused in the paper.

Author Response

(The authors gave the same response as above.)

Reviewer 3 Report

After revision, this paper of Son can be published in Antioxidants. 

Author Response

Thank you for your meaningful review and comments about our paper.

We uploaded the revised manuscript based on your opinion.

I look forward to hearing from you soon.

Thank you very much once again.